# One Solution is Not All You Need:
# Few-Shot Extrapolation via Structured MaxEnt RL

**Saurabh Kumar[1], Aviral Kumar[2], Sergey Levine[2], Chelsea Finn[1]**,
Stanford University[1], UC Berkeley[2]
szk@stanford.edu

## Abstract

While reinforcement learning algorithms can learn effective policies for complex tasks, these policies are often brittle to even minor task variations, especially when variations are not explicitly provided during training. One natural approach to this problem is to train agents with manually specified variation in the training task or environment. However, this may be infeasible in practical situations, either because making perturbations is not possible, or because it is unclear how to choose suitable perturbation strategies without sacrificing performance. The key insight of this work is that learning *diverse* behaviors for accomplishing a task can directly lead to behavior that generalizes to varying environments, without needing to perform explicit perturbations during training. By identifying multiple solutions for the task in a single environment during training, our approach can generalize to new situations by abandoning solutions that are no longer effective and adopting those that are. We theoretically characterize a robustness set of environments that arises from our algorithm and empirically find that our diversity-driven approach can extrapolate to various changes in the environment and task.

## 1   Introduction

Deep reinforcement learning (RL) algorithms have demonstrated promising results on a variety of complex tasks, such as robotic manipulation [22, 13] and strategy games [27, 38]. Yet, these reinforcement learning agents are typically trained in just one environment, leading to performant but narrowly-specialized policies — policies that are optimal under the training conditions, but brittle to even small environment variations [46]. A natural approach to resolving this issue is to simply train the agent on a distribution of environments that correspond to variations of the training environment [4, 9, 18, 33]. These methods assume access to a set of user-specified training environments that capture the properties of the situations that the trained agent will encounter during evaluation. However, this assumption places a significant burden on the user to hand-specify all degrees of variation, or may produce poor generalization along the axes that are not varied sufficiently [46]. Further, varying the environment may not even be possible in the real world.

One way of resolving this problem is to design algorithms that can automatically construct many variants of its training environment and optimize a policy over these variants. One can do so, for example, by training an adversary to perturb the agent [32, 31]. While promising, adversarial optimizations can be brittle, overly pessimistic about the test distribution, and compromise performance. In contrast to both generalization and robustness approaches, humans do not need to practice a task under explicit perturbations in order to adapt to new situations. As a concrete example, consider the task of navigating through a forest with multiple possible paths. Traditional RL approaches may optimize for and memorize the shortest possible path, whereas a person will encounter, *and remember* many different paths during the learning process, including suboptimal paths that still reach the end of the forest. While a single optimal policy would fail if the shortest path becomes unavailable, a repertoire of diverse policies would be robust even when a particular path is no longer

successful (see Figure 1). Concretely, practicing and remembering diverse solutions to a task can naturally lead to robustness. In this work, we consider how we might encourage reinforcement learning agents to do the same – learning a breadth of solutions to a task and remembering those solutions such that they can adaptively switch to a new solution when faced with a new environment.

The key contribution of this work is a framework for policy robustness by optimizing for diversity. Rather than training a single policy to be robust across a distribution over environments, we learn multiple policies such that these behaviors are *collectively* robust to a new distribution over environments. Critically, our approach can be used with only a *single training environment*, rather than requiring access to the entire set of environments over which we wish to generalize. We theoretically characterize the set of environments over which we expect the policies learned by our method to generalize, and empirically find that our approach can learn policies that extrapolate over a variety of aspects of the environment, while also outperforming prior standard and robust reinforcement learning methods.

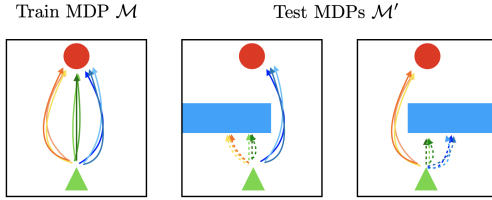

Train MDP $\mathcal{M}$          Test MDPs $\mathcal{M}'$

Figure 1: The key insight of our work is that structured diversity-driven learning in a single training environment (left) can enable few-shot generalization to new environments (right). Our approach learns a parametrized space of diverse policies for solving the training MDP, which enables it to quickly find solutions to new MDPs.

## 2 Preliminaries

The goal in a reinforcement learning problem is to optimize cumulative discounted reward in a Markov decision process (MDP) $\mathcal{M}$, defined by the tuple $(\mathcal{S}, \mathcal{A}, \mathcal{P}, \mathcal{R}, \gamma, \mu)$, where $\mathcal{S}$ is the state space, $\mathcal{A}$ is the action space, $\mathcal{P}(s_{t+1}|s_t, a_t)$ provides the transition dynamics, $\mathcal{R}(s_t, a_t)$ is a reward function, $\gamma$ is a discount factor, and $\mu$ is an initial state distribution. A policy $\pi$ defines a distribution over actions conditioned on the state, $\pi(a_t|s_t)$. Given a policy $\pi$, the probability density function of a particular trajectory $\tau = \{s_i, a_i\}_{i=1}^T$ under policy $\pi$ can be factorized as follows:

$$p(\tau) = \mu(s_0) \cdot \Pi_{t=0}^T \pi(a_t|s_t) \cdot \mathcal{P}(s_{t+1}|s_t, a_t).$$

The expected discounted sum of rewards of a policy $\pi$ is is given by: $R_{\mathcal{M}}(\pi) = \mathbb{E}_{\tau \sim \pi}[R(\tau)] = \mathbb{E} \sum_t \gamma^t \mathcal{R}(s_t, a_t)$. The optimal policy $\pi_{\mathcal{M}}^*$ maximizes the return, $R_{\mathcal{M}}(\pi)$: $\pi_{\mathcal{M}}^* = \arg\max_\pi R_{\mathcal{M}}(\pi)$.

**Latent-Conditioned Policies.** In this work, we will consider policies conditioned on a latent variable. A latent-conditioned policy is described as $\pi(a|s, z)$ and is conditioned on a latent variable $z \in \mathbb{R}^d$. The latent variable $z$ is drawn from a known distribution $z \sim p(Z)$. The probability of observing a trajectory $\tau$ under a latent-conditioned policy is $p(\tau) = \int_z p(\tau|z)p(z)$, where

$$p(\tau|z) = \mu(s_0) \cdot \Pi_{t=0}^T \pi(a_t|s_t, z) \cdot \mathcal{P}(s_{t+1}|s_t, a_t).$$

**Mutual-Information in RL.** In this work, we will maximize the mutual information between trajectories and latent variables. Estimating this quantity is difficult because computing marginal distributions over all possible trajectories, by integrating out $z$, is intractable. We can instead maximize a lower bound on the objective which consists of summing the mutual information between each state $s_t$ in a trajectory $\tau$ and the latent variable $z$. It has been shown that a sum of the mutual information between states in $\tau$, $s_1, \cdots, s_T$, and the latent variable $z$ lower bounds the mutual information $I(\tau, z)$ [19]. Formally, $I(\tau; z) \geq \sum_{t=1}^T I(s_t; z)$.

Finally, we can lower-bound the mutual information between states and latent variables, as $I(S; Z) \geq \mathbb{E}_{z \sim p(z), s \sim \pi(z)}[\log q_\phi(z|s)] - \mathbb{E}_{z \sim p(z)}[\log p(z)]$ [7], where the posterior $p(z|s)$ can be approximated with a learned discriminator $q_\phi(z|s)$.

## 3 Problem Statement: Few-Shot Robustness

In this paper, we aim to learn policies on a single training MDP that can generalize to perturbations of this MDP. In this section, we formalize this intuitive goal into a concrete problem statement that we call "few-shot robustness." During training, the algorithm collects samples from the (single) training MDP, $\mathcal{M} = (\mathcal{S}, \mathcal{A}, \mathcal{P}, \mathcal{R}, \gamma, \mu)$. At test time, the agent is placed in a new test MDP

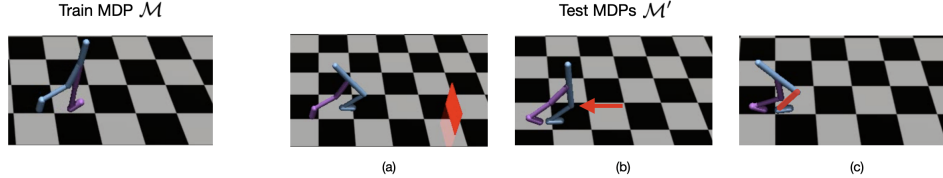

Figure 2: We evaluate SMERL on 3 types of environment perturbations: (a) the presence of an obstacle, (b) a force applied to one of the joints, and (c) motor failure at a subset of the joints.

$\mathcal{M}' = (\mathcal{S}, \mathcal{A}, \mathcal{P}', \mathcal{R}', \gamma, \mu)$, which belongs to a test set of MDPs $S_{\text{test}}$. Each MDP in this test set has identical state and action spaces as $\mathcal{M}$, but may have a different reward and transition function (see Figure 2). In Section 5, we formally define the nature of the changes from training time to test time, which are guided by practical problems of interest, such as the navigation example described in Section 1. In the test MDP, the agent must acquire a policy that is optimal after only a handful of trials. Concretely, we refer to this protocol as *few-shot robustness*, where a trained agent is provided a budget of $k$ episodes of interaction with the test MDP and must return a policy to be evaluated in this MDP. The final policy is evaluated in terms of its expected return in the test MDP $\mathcal{M}'$. Our few-shot robustness protocol at test time resembles the few-shot adaptation performance metric typically used in meta-learning [10], in which a test task is sampled and the performance is measured after allowing the algorithm to adapt to the new test task in a pre-defined budget of $k$ adaptation episodes. While meta-learning algorithms assume access to a distribution of tasks during training, allowing them to benefit from learning the intrinsic structure of this distribution, our setting is more challenging since the algorithm needs to learn from a *single* training MDP only.

## 4   Structured Maximum Entropy Reinforcement Learning

In this section, we present our approach for addressing the few-shot robustness problem defined in Section 3. We first present a concrete optimization problem that optimizes for few-shot robustness, then discuss how to transform this objective into a tractable form, and finally present a practical algorithm. Our algorithm, Structured Maximum Entropy Reinforcement Learning (SMERL), optimizes the approximate objective on a single training MDP.

### 4.1   Optimization with Multiple Policies

Our goal is to be able to learn policies on a single MDP that can achieve (near-)optimal return when executed on a test MDP in the set $S_{\text{test}}$. In order to maximize return on multiple possible test MDPs, we seek to learn a continuous (infinite) subspace or discrete (finite) subset of policies, which we denote as $\bar{\Pi}$. Then, given an MDP $\mathcal{M}' \in S_{\text{test}}$, we select the policy $\pi \in \bar{\Pi}$ that maximizes return $R(\mathcal{M}')$ on the test MDP. We wish to learn $\bar{\Pi}$ such that for any possible test MDP $\mathcal{M}' \in S_{\text{test}}$, there is always an effective policy $\pi \in \bar{\Pi}$. Concretely, this gives rise to our formal training objective:

$$\Pi^* = \arg\max_{\bar{\Pi} \subset \Pi} \left[ \min_{\mathcal{M}' \in S_{\text{test}}} \left( \max_{\pi \in \bar{\Pi}} R_{\mathcal{M}'}(\pi) \right) \right]. \tag{1}$$

Our approach for maximizing the objective in Equation 1 is based on two insights that give rise to a tractable surrogate objective amenable to gradient-based optimization. First, we represent the set $\bar{\Pi}$ using a latent variable policy $\pi(a|s, z)$. Such latent-conditioned policies can express multi-modal distributions. The latent variable can index different policies, making it possible to represent multiple behaviors with a single object. Second, we can produce diverse solutions to a task by encouraging the trajectories of different latent variables $z$ to be distinct while still solving the task. An agent with a repertoire of such *distinct* latent policies can adopt a slightly sub-optimal solution if an optimal policy is no longer viable, or highly sub-optimal, in a test MDP. Concretely, we aim to maximize expected return while also producing unique trajectory distributions.

To encourage distinct trajectories for distinct $z$ values, we introduce a diversity-inducing objective that encourages high mutual information between $p(Z)$, and the marginal trajectory distribution for the latent-conditioned policy $\pi(a|s, z)$. We optimize this objective subject to the constraint that each policy achieves return in $\mathcal{M}$ that is close to the optimal return. This optimization problem is:

$$\theta^* = \arg\max_{\theta} \ I(\tau, z) \ \text{s.t.} \ \forall z, R_{\mathcal{M}}(\pi_{\theta}) \geq R_{\mathcal{M}}(\pi_{\mathcal{M}}^*) - \varepsilon, \tag{2}$$

where $\pi_\theta = \pi_\theta(\cdot|s; z)$ and $\epsilon > 0$. In Section 5, we show that the objective in Equation 2 can be derived as a tractable approximation to Equation 1 under some mild assumptions. The constrained optimization in Equation 2 aims at learning a space of policies, indexed by the latent variable $z$, such that the set $\bar{\Pi} = \{\pi(\cdot|\cdot, z)|z \sim p(z)\}$ covers the space of possible policies $\pi(a|s)$ that induce near-optimal, long-term discounted return on the training MDP $\mathcal{M}$. The mutual information objective $I(\tau, z)$ enforces diversity among policies in $\bar{\Pi}$, but only when these policies are close to optimal.

## 4.2 The SMERL Optimization Problem

In order to tractably solve the optimization problem 2, we lower-bound the mutual information $I(\tau, z)$ by a sum of mutual information terms over individual states appearing in the trajectory $\tau$, as discussed in Section 2. We then obtain the following surrogate, tractable optimization problem:

$$\theta^* = \arg\max_\theta \sum_{t=1}^T I(s_t; z) \text{ s.t. } \forall z, R_\mathcal{M}(\pi_\theta) \geq R_\mathcal{M}(\pi_\mathcal{M}^*) - \varepsilon. \tag{3}$$

Following the argument from [7], we compute an unsupervised reward function from the mutual information between states and latent variables as $\tilde{r}(s, a) = \log q_\phi(z|s) - \log p(z)$, where $q_\phi(z|s)$ is a learned discriminator. Note that $I(s_t; z) = H(z) - H(z|s_t)$, where $H(X)$ is the entropy of random variable $X$. Since the term $H(z)$ encourages the distribution over the latent variables to have high entropy, we fix $p(z)$ to be uniform.

In order to satisfy the constraint in Equation 3 that $\sum_{t=1}^T I(s_t; z)$ is maximized *only when* the latent-conditioned policy achieves return $R_\mathcal{M}(\pi_\theta) \geq R_\mathcal{M}(\pi_\mathcal{M}^*) - \epsilon$, we only optimize the unsupervised reward when the environment return is within a pre-defined $\epsilon$ distance from the optimal return. To this end, we optimize the sum of two quantities: **(1)** the discounted return obtained by executing a latent-conditioned policy in the MDP, $R_\mathcal{M}(\pi_\theta(\cdot|s, z))$, and **(2)** the discounted sum of unsupervised rewards $\sum_t \gamma^t \tilde{r}_t$, only if the policy's return satisfies the condition specified in Equation 3. Combining these components leads to the following optimization in practice ($\mathbb{1}_{[\cdot]}$ is the indicator function, $\alpha > 0$):

$$\theta^* = \arg\max_\theta \mathbb{E}_{z \sim p(z)} \left[ R_\mathcal{M}(\pi_\theta) + \alpha \mathbb{1}_{[R_\mathcal{M}(\pi_\theta) \geq R_\mathcal{M}(\pi_\mathcal{M}^*) - \varepsilon]} \sum_t \gamma^t \tilde{r}(s_t, a_t) \right] \tag{4}$$

## 4.3 Practical Algorithm

We implement SMERL using soft actor-critic (SAC) [16], but with a latent variable maximum entropy policy $\pi_\theta(a|s, z)$. The set of latent variables is chosen to be a fixed discrete set, $Z$, and we set $p(z)$ to be the uniform distribution over this set. At the beginning of each episode, a latent variable $z$ is sampled from $p(z)$ and the policy $\pi_\theta(\cdot|\cdot, z)$ is used to sample a full trajectory, with $z$ being fixed for the entire episode. The transitions obtained, as well as the latent variable $z$, are stored in a replay buffer. When sampling states from the replay buffer, we compute the reward to optimize with SAC according to Equation 3 from Section 4.2:

$$r_{\text{SMERL}}(s_t, a_t) = r(s_t, a_t) + \alpha \mathbb{1}_{R_\mathcal{M}(\pi_\theta) \geq R_\mathcal{M}(\pi_\mathcal{M}^*) - \varepsilon} \tilde{r}(s_t, a_t). \tag{5}$$

For all states sampled from the replay buffer, we optimize the reward obtained from the environment $r$. For states in trajectories which achieve near-optimal return, the agent also receives unsupervised reward $\tilde{r}$, which is higher-valued when the agent visits states that are easy to discriminate, as measured by the likelihood of a discriminator $q_\phi(z|s)$. The discriminator is trained to infer the latent variable $z$ from the states visited when executing that latent-conditioned policy. In order to measure whether $R_\mathcal{M}(\pi_\theta) \geq R_\mathcal{M}(\pi_\mathcal{M}^*) - \varepsilon$, we first train a baseline SAC agent on the environment, and treat the maximum return achieved by the trained SAC agent as the optimal return $R_\mathcal{M}(\pi_\mathcal{M}^*)$. The full training algorithm is described in Algorithm 1.

Following the few-shot robustness evaluation protocol, given a budget of $K$ episodes, each latent variable policy $\pi_\theta(\cdot|s, z)$ is executed in a test MDP $\mathcal{M}'$ for 1 episode. The policy which achieves the maximum sampled return is returned (see Algorithm 2).

# 5 Analysis of Diversity-Driven Learning

We now provide a theoretical analysis of SMERL. We show how the tractable objective shown in Equation 4 can be derived out of the optimization problem in Equation 1 for particular choices of

**Algorithm 1** SMERL: Training in training MDP $\mathcal{M}$

> **while** *not converged* **do**
>     Sample latent $z \sim p(z)$ and initial state $s_0 \sim \mu$.
>     **for** $t \leftarrow 1$ **to** *steps_per_episode* **do**
>         Sample action $a_t \sim \pi_\theta(\cdot|s_t, z_t)$.
>         Step environment: $r_t, s_{t+1} \sim \mathcal{P}(r_t, s_{t+1}|s_t, a_t)$.
>         Compute $q_\phi(z|s_{t+1})$ with discriminator.
>         Let $\tilde{r}_t = \log q_\phi(z|s_{t+1}) - \log p(z)$.
>     Compute $R_\mathcal{M}(\pi_\theta) = \sum_t r_t$.
>     **for** $t \leftarrow 1$ **to** *steps_per_episode* **do**
>         Compute reward $r_{\text{SMERL}}$ according to Eq 5.
>         Update $\theta$ to maximize $r_{\text{SMERL}}$ with SAC.
>         Update $\phi$ to maximize $\log q_\phi(z|s_t)$ with SGD.

**Algorithm 2** SMERL: Few-shot robustness evaluation in test MDP $\mathcal{M}'$

> $R_{\text{MAX}} \leftarrow -\infty$
> $\pi_{\text{best}} \leftarrow \pi_\theta(\cdot|z_0)$
> **for** $i \in \{1, 2, ..., K\}$ **do**
>     Rollout policy $\pi_\theta(\cdot|z_i)$ in MDP $\mathcal{M}'$ for
>     1 episode and compute $R_{\mathcal{M}'}(\pi_\theta)$.
>     $R_{\text{MAX}} \leftarrow \max(R_{\text{MAX}}, R_{\mathcal{M}'}(\pi_\theta))$
>     Update $\pi_{\text{best}}$
> Return $\pi_{\text{best}}$

robustness sets $S_{\text{test}}$ of MDPs. Our analysis in divided into three parts. First, we define our choice of MDP robustness set. We then provide a reduction of this set over MDPs to a robustness set over policies. Finally, we show that an optimal solution of our tractable objective is indeed optimal for this policy robustness set under certain assumptions.

## 5.1 Robustness Sets of MDPs and Policies

Following our problem definition in Section 2, our robustness sets will be defined over MDPs $\mathcal{M}'$, which correspond to versions of the training MDP $\mathcal{M}$ with altered reward or dynamics. For the purpose of this discussion, we limit ourselves to discrete state and action spaces. Drawing inspiration from the navigation example in Section 1, we now define our choice of robustness set, which we will later connect to the set of MDPs to which we can expect SMERL to generalize. Hence, we define the MDP robustness set as:

**Definition 1.** *Given a training MDP $\mathcal{M}$ and $\epsilon > 0$, the MDP robustness set, $S_{\mathcal{M},\epsilon}$, is the set of all MDPs $\mathcal{M}'$ which satisfy two properties:*

$$(1) R_\mathcal{M}(\pi_\mathcal{M}^*) - R_\mathcal{M}(\pi_{\mathcal{M}'}^*) \leq \epsilon$$

*(2) The trajectory distribution of the policy $\pi_{\mathcal{M}'}^*$ is the same on $\mathcal{M}$ and $\mathcal{M}'$.*

Intuitively, the set $S_{\mathcal{M},\epsilon}$ consists of all MDPs for which the optimal policy $\pi_{\mathcal{M}'}^*$ achieves a return *on the training MDP* that is close to the return achieved by its own optimal policy, $\pi_\mathcal{M}^*$. Additionally, the optimal policy of $\mathcal{M}'$ must produce the same trajectory distribution in $\mathcal{M}'$ as in $\mathcal{M}$. These properties are motivated by practical situations, where a perturbation to a training MDP, such as an obstacle blocking an agent's path (see Figure 1), allows different policies in the training MDP to be optimal on the test MDP. This perturbation creates a test MDP $\mathcal{M}'$ whose optimal policy achieves return close to the optimal policy of $\mathcal{M}$ since it takes only a slightly longer path to the goal, and that path is traversed by the same policy in the original MDP $\mathcal{M}$. Given this intuition, the MDP robustness set $S_{\mathcal{M},\epsilon}$ will be the set that we use for the test set of MDPs $S_{\text{test}}$ in Equation 1 in our upcoming derivation.

While we wish to generalize to MDPs in the MDP robustness set, in our training protocol an RL agent has access to only a single training MDP. It is not possible directly optimize over the set of test MDPs, and SMERL instead optimizes over policies in the training MDP. In order to analyze the connection between the policies learned by SMERL and robustness to test MDPs, we consider a related robustness set, defined in terms of sub-optimal policies on the training MDP:

**Definition 2.** *Given a training MDP $\mathcal{M}$ and $\epsilon > 0$, the policy robustness set, $S_{\pi_\mathcal{M}^*,\epsilon}$ is defined as*

$$S_{\pi_\mathcal{M}^*,\epsilon} = \{\pi \mid R_\mathcal{M}(\pi_\mathcal{M}^*) - R_\mathcal{M}(\pi) \leq \epsilon \text{ and } \pi \text{ is a deterministic policy }\}.$$

The policy robustness set consists of all policies which achieve return close to the optimal return of the training MDP. Since the optimal policies of the MDP robustness set also satisfy this condition, intuitively, $S_{\pi_\mathcal{M}^*,\epsilon}$ encompasses the optimal policies for MDPs from $S_{\mathcal{M},\epsilon}$.

Next, we formalize this intuition, and in Sec. 5.3 show how this convenient relationship can replace the optimization over $S_{\mathcal{M},\epsilon}$ in Eq. 1 with an optimization over policies, as performed by SMERL.

## 5.2 Connecting MDP Robustness Sets with Policy Robustness Sets

Every policy in $S_{\pi_\mathcal{M}^*,\epsilon}$ is optimal for some MDP in $S_{\mathcal{M},\epsilon}$. Thus, if an agent can learn all policies in $S_{\pi_\mathcal{M}^*,\epsilon}$, then we can guarantee the ability to perform optimally in each and every possible MDP that

can be encountered at test time. In order to formally prove this intuition, we provide a set of two containment results. Proofs from this section can be found in Appendix A.

**Proposition 1.** *For each MDP $\mathcal{M}'$ in the MDP robustness set $S_{\mathcal{M},\epsilon}$, $\pi^*_{\mathcal{M}'}$ exists in the policy robustness set $S_{\pi^*_{\mathcal{M}},\epsilon}$.*

**Proposition 2.** *Given an MDP $\mathcal{M}$ and each policy $\pi$ in the policy robustness set $S_{\pi^*_{\mathcal{M}},\epsilon}$, there exists an MDP $\mathcal{M}' = (\mathcal{S}, \mathcal{A}, \bar{\mathcal{P}}, \bar{\mathcal{R}}, \gamma, \mu)$ such that $\mathcal{M}' \in S_{\mathcal{M},\epsilon}$ and $\pi = \pi^*_{\mathcal{M}'}$.*

We next use this connection between $S_{\pi^*_{\mathcal{M}},\epsilon}$ an $S_{\mathcal{M},\epsilon}$ to verify that SMERL indeed finds a solution to our formal training objective (Equation 1).

### 5.3 Optimizing the Robustness Objective

Now that we have shown that any policy in $S_{\pi^*_{\mathcal{M}},\epsilon}$ is optimal in some MDP in $S_{\mathcal{M},\epsilon}$, we now show how this relation can be utilized to simplify the objective in Equation 1. Finally, we show that this simplification naturally leads to the trajectory-centric mutual information objective. We first introduce a modified training objective below in Equation 6, and then show in Proposition 3 that under some mild conditions, the solution obtained by optimizing Equation 6 matches the solution obtained by solving Equation 1:

$$\Pi^* = \arg\max_{\bar{\Pi} \subset \Pi} \min_{\hat{\pi} \in S_{\pi^*_{\mathcal{M}},\epsilon}} [\max_{\pi \in \bar{\Pi}} \mathbb{E}_{\tau \sim \hat{\pi}} \log p(\tau|\pi)]. \tag{6}$$

**Proposition 3.** *The solution to the objective in Equation 1 is the same as the solution to the objective in Equation 6 when $S_{test} = S_{\mathcal{M},\epsilon}$.*

Finally, we now show that the set of policies obtained by optimizing Equation 6 is the same as the set of solutions obtained by the SMERL mutual information objective (Equation 2).

**Proposition 4.** *[Informal] With usual notation and for a sufficiently large number of latent variables, the set of policies $\Pi^*$ that result from solving the optimization problem in Equation 6 is equivalent to the set of policies $\pi_{\theta^*,z}$ that result from solving the optimization problem in Equation 2.*

A more formal theorem statement and its proof are in Appendix A. Propositions 3 and 4 connect the solutions to the optimization problems in Equation 1 and Equation 2, for a specific instantiation of $S_{\text{test}}$. Our results in this section suggest that the general paradigm of diversity-driven learning is effective for robustness when the test MDPs satisfy certain properties. In Section 7, we will empirically measure SMERL's robustness when optimizing the SMERL objective on practical problems where the test perturbations satisfy these conditions.

## 6 Related Work

Our work is at the intersection of robust reinforcement learning methods and reinforcement learning methods that promote generalization, both of which we review here. Robustness is a long-studied topic in control and reinforcement learning [47, 29, 44] in fields such as robust control, Bayesian reinforcement learning, and risk-sensitive RL [5, 3]. Works in these areas typically focus on linear systems or finite MDPs, while we aim to study high-dimensional continuous control tasks with complex non-linear dynamics. Recent works have aimed to bring this rich body of work to modern deep reinforcement learning algorithms by using ensembles of models [33, 20], distributions over critics [39, 2], or surrogate reward estimation [43] to represent and reason about uncertainty. These methods assume that the conditions encountered during training are representative of those during testing, an assumption also common in works that study generalization in reinforcement learning [4, 18] and domain randomization [34, 41] . We instead focus specifically on extrapolation, and develop an algorithm that generalizes to new, out-of-distribution dynamics after training in a single MDP.

Other works have noted the susceptibility of deep policies to adversarial attacks [17, 25, 31, 11]. Unlike these works, we focus on generalization to new environments, rather than robustness in the presence of an adversary. Nevertheless, a number of prior works have considered worst-case formulations to robustness by introducing such an adversary that can perturb the agent [32, 31, 23, 30, 40], which can promote generalization. In contrast, our formulation does not require explicit perturbations during training, nor an adversarial optimization.

Our approach specifically enables an agent to adapt to an out-of-distribution MDP by searching over a latent space of skills. We use latent-conditioned policies rather than sampling policy parameters

with a hyperpolicy [36], since it is easier for the discriminator in SMERL to predict a latent variable rather than the parameters of the policy. Our formulation is similar to prior work which maximizes the mutual information between a distribution of latent conditioned policies and the latent variable on which they condition [1]. In contrast to this prior work, SMERL maximizes the mutual information between states and latent variables $z$, and includes a threshold to toggle the mutual information component of the objective. Our method of searching over a latent space of skills resembles approaches that search over models via system identification [45] or quickly adapt to new MDPs via meta-learning [10, 6, 28, 15, 35, 21, 8]. In contrast, we do not require a distribution over dynamics or tasks during training. Instead, we drive the agent to identify diverse solutions to a single task. To that end, our derivation draws similarities to prior works on unsupervised skill discovery [7, 37, 19, 14, 12]. It is also similar to works which learn latent conditioned policies for diverse behaviors in imitation learning [24, 26]. These works are orthogonal and complementary to our work, as we focus on and formalize how such approaches can be leveraged in combination with task rewards to achieve robustness.

# 7 Experimental Evaluation

The goal of our experimental evaluation is to test the central hypothesis of our work: does structured diversity-driven learning lead to policies that generalize to new MDPs? We also qualitatively study the behaviors produced by our approach and compare the performance of our method relative to prior approaches for generalizable and robust policy learning. To this end, we conduct experiments within both an illustrative 2D navigation environment with a point mass and three continuous control environments using the MuJoCo physics engine [42]: HalfCheetah-Goal, Walker2d-Velocity, and Hopper-Velocity.

The state space of the 2D navigation environment is a 4x4 arena, and the agent can take actions in a 2-dimensional action space to move its position. The agent begins in the bottom left corner and its task is to navigate to a goal position in the upper right corner. In HalfCheetah-Goal, the task is to navigate to a target goal location. In Walker-Velocity and Hopper-Velocity, the task is to move forward at a particular velocity. We perform evaluation in three types of test conditions: (1) an obstacle is present on the path to the goal, (2) a force is applied to one of the joints at a pre-specified small time interval $(t_1, t_2)$ from time step $t_1 = 10$ to time step $t_2 = 15$, and (3) a subset of the motors fail for time intervals of varying lengths. With a small perturbation magnitude (e.g. short obstacle height or small magnitude of force), the optimal policies achieve near-optimal return when executed on the training MDP, and these policies yield the same trajectories when executed on the train and test MDPs, satisfying the definition of our MDP robustness set (see Definition 1) which we can expect SMERL to be robust to. When large perturbation magnitude, the first condition is no longer satisfied.

For each test environment, we vary the amount of perturbation to measure the degree to which different algorithms are robust. We vary the height of the obstacle in (1), the magnitude of the force in (2), and the number of time steps for which motor failure occurs in (3). Further environment specifications, such as the start and goal positions, the target velocity value, and the exact form of the reward function, are detailed in Appendix B.

## 7.1 What Policies Does SMERL Learn?

We first study the policies learned by SMERL on a point mass navigation task. With SAC and SMERL, we learn 6 latent policies. As shown in Figure 3, SMERL produces latent-conditioned policies that solve the task by taking distinct paths to the goal, and where trajectories from the same latent (shown in the same color) are consistent. This collection of policies provides robustness to environment perturbations. In contrast, nearly all trajectories produced by SAC follow a straight path to the goal, which may become inaccessible in new environments.

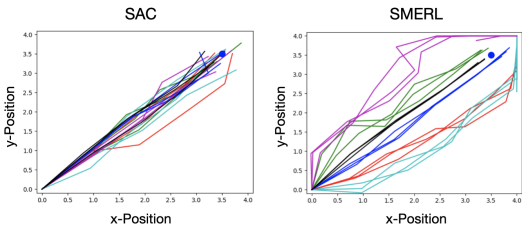

Figure 3: Trajectories produced by SAC and SMERL on the 2D goal navigation environment. The task is to navigate to within a 0.5 radius of a goal position, indicated with a blue dot. Trajectories associated with different latent-conditioned policies are are illustrated using different colors.

## 7.2 Can SMERL Quickly Generalize To Extrapolated Environments?

Given that SMERL learns distinguishable and diverse policies in simple environments, we now study whether these policies are robust to various test conditions in more challenging continuous-control

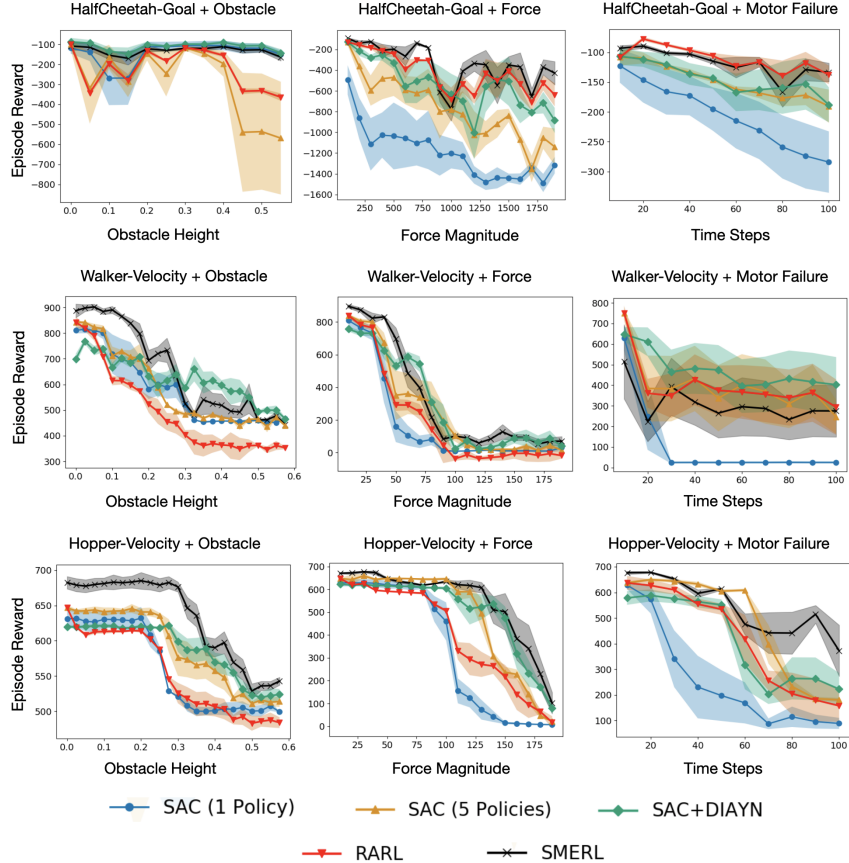

Figure 4: We compare the robustness of SAC with 1 Policy, SAC with 5 Policies, SAC+DIAYN, RARL, and SMERL on 3 types of perturbations, on HalfCheetah-Goal Walker-Velocity, and Hopper-Velocity. SMERL is more consistently robust to environment perturbations than other maximum entropy, diversity seeking, and robust RL methods. We plot the mean across 3 seeds for all test environments. The shaded region is 0.5 standard deviation below to 0.5 standard deviation above the mean.

problems. We compare SMERL to standard maximum-entropy RL (SAC), an approach that learns multiple diverse policies but does not maximize a reward signal from the environment (DIAYN), a naive combination of SAC and DIAYN (SAC+DIAYN), and Robust Adversarial Reinforcement Learning (RARL), which is a robust RL method. In SAC+DIAYN, the unsupervised DIAYN reward and task reward are added together. By comparing to SAC and DIAYN, we aim to test the importance of learning diverse policies and of ensuring near-optimal return, for achieving robustness. By comparing to RARL, we aim to understand how SMERL compares to adversarial optimization approaches.

For SMERL, SAC, DIAYN, and SAC+DIAYN, we learn $|Z| = 5$ latent-conditioned policies, and follow our few-shot evaluation protocol described in Section 3 with budget $k = 5$. We also compare to SAC with $|Z| = 1$ (reported as "SAC (1 Policy)" in Figure 4). Specifically, we run every latent-conditioned policy once and select the policy which achieves the highest return on that environment; we then run the selected policies 5 times and compute the averaged performance. For more details, including a description of how hyperparameters are selected and a hyperparameter sensitivity analysis, see Appendix B.

We report results in Figure 4. On all three environments and all three types of test perturbations, we find that SMERL is consistently as robust or more robust than SAC (1 Policy) and SAC (5 Policies). Interestingly, even when the perturbation amount is small, SMERL outperforms SAC. As the perturbation magnitude increases (e.g. obstacle height, force magnitude, or agent's mass), the performance of SAC quickly drops. SMERL's performance also drops, but to a lesser degree. With large perturbation magnitudes, all methods fail to complete the task on most test environments, as we expect. Interestingly, SAC with 5 latent-conditioned policies outperforms SAC with a single

| Force Magnitude | Policy 1 | Policy 2 | Policy 3 | Policy 4 | Policy 5 |
|---|---|---|---|---|---|
| 0.0 | -86.3 | -87.2 | -133.1 | -77.0 | **-72.3** |
| 100.0 | -88.9 | -92.8 | -87.5 | -107.8 | **-83.8** |
| 300.0 | **-222.7** | -357.0 | -397.9 | -1238.7 | -424.1 |
| 500.0 | -868.9 | -528.3 | **-283.7** | -1196.3 | -669.5 |
| 700.0 | -1046.6 | -951.5 | -769.7 | -1758.8 | **-913.9** |
| 900.0 | -1249.3 | **-1238.3** | -1425.1 | -1264.4 | -1282.5 |

Table 1: SMERL policy performance and selection on HalfCheetah-Goal+Force test environments.

latent-conditioned policy on all test environments with the exception of HalfCheetah-Goal + Obstacle, indicating that learning multiple policies is beneficial. However, SMERL's improvement over SAC (5 policies) highlights the importance of learning diverse solutions to the task, in addition to simply having multiple policies. RARL, which also only has a single policy, is more robust than SAC (1 policy) to the force and motor failure perturbations but is less robust when an obstacle is present.

DIAYN learns multiple diverse policies, but since it is trained independently of task reward, it only occasionally solves the task and otherwise produces policies that perform structured diverse behavior but do not achieve near-optimal return. For clarity, we omit DIAYN results from Figure 4. For a comparison with DIAYN, see Figure 5 in Appendix B. The performance of SAC+DIAYN is worse than or comparable to SMERL, with the exception of the Walker-Velocity + Motor Failure test environments and to some degree on Walker-Velocity + Obstacle. This suggests that naively summing the environment and unsupervised reward can achieve some degree of robustness, but is not consistent. In contrast, SMERL balances the task reward and DIAYN reward to achieve few-shot robustness, since it only adds the DIAYN reward when the latent policies are near-optimal.

### 7.3 Does SMERL Select Different Policies At Test Time?

We analyze the individual policy performance of SMERL on a subset of the HalfCheetah-Goal + Force test environments to understand how much variation there is among the performance of different policies and the correlation between train performance and test performance of each policy (see Table 1.When the force magnitude is 0.0 and 100.0, the variation in performance between policies is relatively small (max difference in performance is 24.2 when the force magnitude is 100.0) as compared to higher magnitudes (the max difference is 1016 with the force magnitude is 300.0). Additionally, high train performance doesn't necessarily correlate with high test performance. For example, policy 5 performs the best on the train environment (force magnitude 0.0) but is the second worst among the 5 policies when the force magnitude is 300.0. These results indicate that there is not a single policy that performs best on all test environments, so having multiple diverse policies provides alternatives when a particular policy becomes highly sub-optimal. For a more complete set of results on how SMERL selects policies on the test environments for HalfCheetah-Goal, Walker-Velocity, and Hopper-Velocity, see See Appendix B.4.

## 8 Conclusion

In this paper, we present a robust RL algorithm, SMERL, for learning RL policies that can extrapolate to out-of-distribution test conditions with only a small number of trials. The core idea underlying SMERL is that we can learn a set of policies that finds multiple diverse solutions to a single task. In our theoretical analysis of SMERL, we formally describe the types of test MDPs under which we can expect SMERL to generalize. Our empirical results suggest that SMERL is more robust to various test conditions and outperforms prior diversity-driven RL approaches.

There are several potential future directions of research to further extend and develop the approach of structured maximum entropy RL. One promising direction would be to develop a more sophisticated test-time adaptation mechanism than enumerated trial-and-error, e.g. by using first-order optimization. Additionally, while our approach learns multiple policies, it leaves open the question of how many policies are necessary for different situations. We may be able to address this challenge by learning a policy conditioned on a *continuous* latent variable, rather than a finite number of behaviors. Finally, structured max-ent RL may be helpful for situations other than robustness, such as hierarchical RL or transfer learning settings when learned behaviors need to be reused for new purposes. We leave this as an exciting direction for future work.

## Broader Impacts

### Applications and Benefits

Our diversity-driven learning approach for improved robustness can be beneficial for bringing RL to real-world applications, such as robotics. It is critical that various types of robots, including service robotics, home robots, and robots used for disaster relief or search-and-rescue are able to handle varying environment conditions. Otherwise, they may fail to complete the tasks they are supposed to accomplish, which could have significant consequences in safety-critical situations.

It is conceivable that, during deployment of robotics systems, the system may encounter changes in its environment that it has not previously dealt with. For example, a robot may be tasked with picking up a set of objects. At test time, the environment may slightly differ from the training setting, e.g. some objects may be missing or additional objects may be present. These previously unseen configurations may confuse the agent's policy and lead to unpredictable and sub-optimal behavior. If RL algorithms are to be used to prescribe actions from input observations in a robotics application, the algorithms must be robust to these perturbations. Our approach of learning multiple diverse solutions to the task is a step towards achieving the desired robustness.

### Risks and Ethical Issues

RL algorithms, in general, face a number of risks. First, they tend to suffer from reward specification - in particular, the reward may not necessarily be completely aligned with the desired behavior. Therefore, it can be difficult for a practitioner to predict the behavior of an algorithm when it is deployed. Since our algorithm learns multiple ways to optimize a task reward, the robustness and predictability of its behavior is also limited by the alignment of the reward function with the qualitative task objective. Additionally, even if the reward is well-specified, RL algorithms face a number of other risks, including (but not limited to) safety and stability. Our diversity-driven learning paradigm suffers from the same issues, as different latent-conditioned policies may not produce reliable behavior when executed in real world settings if the underlying RL algorithm is unstable.

## Acknowledgements and Disclosure of Funding

We thank Kyle Hsu and Benjamin Eysenbach for sharing implementations of DIAYN, and Abhishek Gupta for helpful discussions. We thank Eric Mitchell, Ben Eysenbach, and Justin Fu for their feedback on an earlier version of this paper. Saurabh Kumar is supported by an NSF Graduate Research Fellowship and the Stanford Knight Hennessy Fellowship. Aviral Kumar is supported by the DARPA Assured Autonomy Program. Chelsea Finn is a CIFAR Fellow in the Learning and Machines and Brains program.

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
