[Supplementary Material]

# Appendices

## A  Proofs

**Proposition 1.** *For each MDP $\mathcal{M}'$ in the MDP robustness set $S_{\mathcal{M},\epsilon}$, $\pi^*_{\mathcal{M}'}$ exists in the policy robustness set $S_{\pi^*_{\mathcal{M}},\epsilon}$.*

*Proof.* This result follows by the definition of the set $S_{\mathcal{M},\epsilon}$. $\qquad\square$

**Proposition 2.** *Given an MDP $\mathcal{M}$ and each policy $\pi$ in the policy robustness set $S_{\pi^*_{\mathcal{M}},\epsilon}$, there exists an MDP $\mathcal{M}' = (\mathcal{S}, \mathcal{A}, \bar{\mathcal{P}}, \bar{\mathcal{R}}, \gamma, \mu)$ such that $\mathcal{M}' \in S_{\mathcal{M},\epsilon}$ and $\pi = \pi^*_{\mathcal{M}'}$.*

*Proof.* This argument can be shown by first noting that the value of any policy, $\pi$, in an MDP can be written as, $Q^\pi(P) = (I - \gamma P^\pi)^{-1}[R]$. Now, for any given policy $\pi \in S_{\pi^*_{\mathcal{M}},\epsilon}$, we show that we can modify the dynamics to $P'$ such that $Q^\pi(P') \geq Q^{\pi'}(P')$, for all other policies $\pi'$. Such a dynamics $P'$ always exists for any policy $\pi$, since for any optimal policy $\pi'$ in the original MDP with transition dynamics $P$, we can re-write $P \cdot \pi$ as $P \cdot \pi' = P \cdot \pi' \cdot \frac{\pi}{\pi'}$ and by modifying the transition dynamics, as $P' = P \cdot \frac{\pi}{\pi'}$. With this transformation, $\pi'$ is optimal in this modified MDP with dynamics $P'$. $\qquad\square$

**Proposition 3.** *The solution to the objective in Equation 1 is the same as the solution to the objective in Equation 6 when $S_{test} = S_{\mathcal{M},\epsilon}$.*

*Proof.* We first note that for any optimal policy $\pi^*_{\mathcal{M}'}$ of an MDP $\mathcal{M}' \in S_{\mathcal{M},\epsilon}$, the trajectory distribution in the original MDP, $p_{\mathcal{M}}(\tau|\pi^*_{\mathcal{M}'})$, is the same as the trajectory distribution in the perturbed MDP, $\mathcal{M}'$, $p_{\mathcal{M}'}(\tau|\pi^*_{\mathcal{M}'})$, due to the definition of $S_{\mathcal{M},\epsilon}$. Formally,

$$\forall \mathcal{M}' \in S_{\mathcal{M},\epsilon}, \;\; p_{\mathcal{M}'}(\tau|\pi^*_{\mathcal{M}'}) = p_{\mathcal{M}}(\tau|\pi^*_{\mathcal{M}'}).$$

Thus our problem reduces to learning a policy $\pi$ that attains the same trajectory distribution as $\pi^*_{\mathcal{M}'}$ in MDP $\mathcal{M}'$, which is also the trajectory distribution of $\pi^*_{\mathcal{M}'}$ in $\mathcal{M}$. Further, we know that the policy $\pi^*_{\mathcal{M}'}$ is contained in the policy robustness set, $S_{\pi^*_{\mathcal{M}},\epsilon}$, hence, there exists at least one policy in set $\bar{\Pi}$, that generates the same trajectory distribution, and as a result, maximizes the expected likelihood, $\max_{\pi \in \bar{\Pi}} E_{\tau \sim \pi'}[\log p(\tau|\pi)]$, for any policy $\pi' \in S_{\pi^*_{\mathcal{M}},\epsilon}$. We call this "trajectory distribution matching."

The objective in Equation 6 precisely uses this connection – it searches for a set of policies, $\bar{\Pi}$, such that at least one policy $\pi' \in \bar{\Pi}$ maximizes the expected log-likelihood of trajectory distribution, i.e. matches the trajectory distribution, of any given policy $\pi^*_{\mathcal{M}'} \in S_{\pi^*_{\mathcal{M}},\epsilon}$, which is identical to the set of optimal policies for $\mathcal{M}' \in S_{\mathcal{M},\epsilon}$. Moreover, this likelihood-based "trajectory matching" can be performed directly in the original MDP, $\mathcal{M}$, since optimal policies for $\mathcal{M}'$ admit the same trajectory distribution in both $\mathcal{M}$ and $\mathcal{M}'$, hence proving the desired result. $\qquad\square$

**Proposition 4.** *[Informal] With usual notation and for a sufficiently large number of latent variables, the set of policies $\Pi^*$ that result from solving the optimization problem in Equation 6 is equivalent to the set of policies $\pi_{\theta^*,z}$ that result from solving the optimization problem in Equation 2.*

We formalize this statement as follows:
With usual notation and for $|Z| = |\bar{\Pi}| \geq |S_{\pi^*_{\mathcal{M}},\epsilon}|$, $\Pi^* = \{\pi_{\theta^*,z}\}$.

*Proof.* Given that $|\bar{\Pi}| \geq |S_{\pi^*_{\mathcal{M}},\epsilon}|$, we can rewrite the optimization problem in Equation 6 as:

$$\min_{\hat{\pi} \in S_{\pi^*_{\mathcal{M}},\epsilon}} [\max_{\pi \in \bar{\Pi}} \mathbb{E}_{\tau \sim \hat{\pi}} \log p(\tau|\pi)]. \tag{7}$$

Note that under deterministic dynamics, we have:

$\mathbb{E}_{\tau \sim \hat{\pi}} \log p(\tau|\pi) = \sum \mathbb{E}_{a_t \sim \hat{\pi}}[\log \pi(a_t|\pi)].$

Let $p(x) = \hat{\pi}$, and let $q(x) = \pi$. Then, we have:

$$\min_{p(x)} \max_{q(x)} \mathbb{E}_{x \sim p(x)}[\log q(x)]$$

We know that

$$\mathbb{E}_{x \sim p(x)}[\log p(x) - \log q(x)] \geq 0.$$

This implies that:

$$\max_{q(x)} \mathbb{E}_{x \sim p(x)}[\log q(x)] = -\mathcal{H}(p(x)).$$

where $\mathcal{H}$ is Shannon entropy.

Hence,

$$\min_{p(x)} \max_{q(x)} \mathbb{E}_{x \sim p(x)}[\log q(x)] = \min_{p(x)} -\mathcal{H}(p(x)) = \max_{p(x)} \mathcal{H}(p(x))$$

$$= \max_{q(x)} \mathcal{H}(q(x)) = \max_{\pi} \mathcal{H}(p(\tau|\pi)) = \max_{\pi} I(\tau, z).$$

where the last equality holds since $\mathcal{H}(\tau|z) = 0$ under our assumption that $|\bar{\Pi}|$ is sufficiently large.

$\square$

**Remark 1.** *When $|Z| < |S_{\pi^*_{\mathcal{M}}, \epsilon}|$, the conditional entropy $H(\tau|z)$ is non-zero, so we constrain the conditional entropy to be small. This results in maximizing $H(\tau)$ and minimizing $H(\tau|z)$ which overall maximizes the mutual information $I(\tau, z)$.*

**Remark 2.** *When $|Z| < |S_{\pi^*_{\mathcal{M}}, \epsilon}|$, we require a metric defined on the space of trajectories in order to quantify how much better it is to choose one policy with respect other policies in the set of latent-conditioned policies.*

## B   Experimental Setup and Additional Results

For all experiments, we used a NVIDIA TITAN RTX GPU. SAC and SMERL train in 25 minutes on the 2D Navigation task. All agents train in 6 hours on the Walker-Velocity and Hopper-Velocity environments. All agents train in 1.5 hours on HalfCheetah-Goal.

### B.1   Environments

In the 2D navigation environment, the reward function is the negative distance to the goal position. The agent begins at $(x, y) = (0, 0)$, and the goal position is at $(3.5, 3.5)$.

In HalfCheetah-Goal, the goal location is at $(3.0, 0.0)$, where the first coordinate is the x-position and the second coordinate is the y-position. The reward function is the negative absolute value distance to the goal, computed by subtracting the x-position of the goal from the x-position of the agent. The max episode length is 500 time steps. In Walker-Velocity and Hopper-Velocity, the target velocity is 5.0. The reward function adds $\min(velocity_t, 5.0)$ to the original reward functions $r_t$ of Walker / Hopper, where $velocity_t$ is the agent's velocity at the current time step $t$. The max episode length is 200 time steps. In all environments, the agent's starting position is at $(0, 0)$.

The test perturbations are constructed as follows. For the obstacle perturbation, an obstacle is present at $(0.001, 0.0)$ for HalfCheetah-Goal + Obstacle, and at $(2.5, 0.0)$ for Walker-Velocity and Hopper-Velocity. Each obstacle test environment has an obstacle with a different height, and the obstacle heights vary from 0.0 to 0.6. For the force perturbation, a negative force is applied at the fifth joint of the cheetah, walker, and hopper agents, from time step $t_1 = 10$ to $t_2 = 15$. Each force test environment has a different force amount applied, and the force varies from 0 to $-1900$ for the HalfCheetah-Goal + Force test environments, and it varies from 0 to $-190$ for the Walker-Velocity + Force and Hopper-Velocity + Force test environments. For the motor failure perturbation, actions 0, 1, 3, and 4 are zeroed out for the cheetah agent, actions 0 and 1 for the walker agent, and actions 0 and 1 for the hopper agent, from time steps 10 to $10 + t$. Each test environment has a unique $t$, where $t$ varies from 0 to 100.

Figure 5: Figure comparing the robustness of SAC with 1 Policy, SAC with 5 Policies, DIAYN, SAC+DIAYN, RARL, and SMERL on 3 types of perturbations, on HalfCheetah-Goal Walker-Velocity, and Hopper-Velocity. SMERL is more consistently robust to environment perturbations than other maximum entropy, diversity seeking, and robust RL methods. We plot the mean across 3 seeds for all test environments. The shaded region is 0.5 standard deviation below to 0.5 standard deviation above the mean.

## B.2 Hyperparameters

Table 3 lists the common SAC parameters used in the comparative evaluation in Figure 4, as well as the values of $\alpha$ and $\epsilon$ used in SMERL. While the policies learned on the train MDP are stochastic, during evaluation, we select the mean action (SAC, DIAYN, and SMERL). This does not make the performance worse for any of the approaches.

To estimate the optimal return value that SMERL requires, we trained SAC on a single training environment using 1 seed. We used the final SAC performancee (the SAC return $R_{\text{SAC}}$) on each train environment as the optimal return value estimate for that environment. We then selected a value of $\epsilon$ for SMERL to set a return threshold $R_{\text{SAC}} - \epsilon$ above which the unsupervised reward is added, and a value of $\alpha$ which weights the unsupervised reward. To select $\epsilon$ and $\alpha$, we trained SMERL agents with a single seed on HalfCheetah-Goal using $\alpha \in \{0.1, 0.5, 1.0, 10.0\}$ and $\epsilon \in \{0.1R_{\text{SAC}}, 0.2R_{\text{SAC}}, 0.3R_{\text{SAC}}\}$, and evaluated their performance on a single HalfCheetah-Goal + Obstacle (obstacle height = 0.2). We used the same protocol to select $\alpha$ for SAC+DIAYN. We found $\alpha = 10.0$ and $\epsilon = 0.1R_{\text{SAC}}$ to work best for SMERL and $\alpha = 0.5$ to work best for SAC+DIAYN. We used these values when training SMERL and SAC+DIAYN and evaluating on all test environments.

RARL required more data to reach the same level of performance as a fully-trained SAC agent on each training environment, so we trained RARL for $5\times$ the number of environment steps as SAC, SMERL, DIAYN, and SAC+DIAYN. RARL trains a model-free RL agent jointly with an adversary which perturbs the agent's actions. We train the adversary to apply 2D forces on the torso and feet

| Hyperparameters for 2D Navigation experiment | |
|---|---|
| Parameter | Value |
| optimizer | Adam |
| learning rate | $3 \cdot 10^{-4}$ |
| discount ($\gamma$) | 0.99 |
| replay buffer size | $10^3$ |
| number of hidden layers | 2 |
| number of hidden units per layer | 32 |
| number of samples per minibatch | 128 |
| nonlinearity | RELU |
| target smoothing coefficient ($\tau$) | 0.01 |
| target update interval | 1 |
| gradient steps | 1 |
| SMERL: value of $\alpha$ | 10.0 |
| SMERL: value of $\epsilon$ | $0.05 R_{\text{SAC}}$ |

Table 2: Hyperparameters used for SAC and SMERL for the 2D navigation experiment.

| Hyperparameters for continuous control experiments | |
|---|---|
| Parameter | Value |
| optimizer | Adam |
| learning rate | $3 \cdot 10^{-4}$ |
| discount ($\gamma$) | 0.99 |
| replay buffer size | $10^3$ |
| number of hidden layers | 2 |
| number of hidden units per layer | 256 |
| number of samples per minibatch | 256 |
| nonlinearity | RELU |
| target smoothing coefficient ($\tau$) | 0.005 |
| target update interval | 1 |
| gradient steps | 1 |
| SMERL: value of $\alpha$ | 10.0 |
| SMERL: value of $\epsilon$ | $0.1 R_{\text{SAC}}$ |

Table 3: Hyperparameters used for SAC, DIAYN, SAC+DIAYN, and SMERL for continuous control experiments.

of the cheetah, walker, and hopper in HalfCheetah-Goal, Walker-Velocity, and Hopper-Velocity, respectively, following the same protocol as done by the authors [32]. Hyperparameters of TRPO, the policy optimizer for the protagonist and adversarial policies in RARL, are selected by grid search on HalfCheetah-Goal, evaluating performance on one HalfCheetah-Goal + Obstacle test environment (obstacle height = 0.2). These hyperparameters were then kept fixed for all experiments on HalfCheetah-Goal, Walker-Velocity, and Hopper-Velocity.

### B.3 Hyperparameter Sensitivity Analysis

We also a perform a more detailed hyperparameter sensitivity analysis for SMERL and SAC+DIAYN. On HalfCheetah-Goal, we examine the effect of varying $\epsilon$ and $\alpha$ on the evaluation performance of SMERL (see Figure 6. We perform this hyperparameter study on two test environments: HalfCheetah-Goal + Obstacle and HalfCheetah-Goal + Force. We find that the robustness of SMERL is sensitive to the choice of $\epsilon$: $\epsilon = 0.1R$ results in policies that are more robust to the environment perturbations, and $\alpha = 10.0$ generally works best.

On HalfCheetah-Goal and WalkerVelocity, we also examine the effect of varying $\alpha$, the weight by which the unsupervised reward is multiplied, on the evaluation performance of SAC+DIAYN (see Figure 7). We find that the robustness of SAC+DIAYN is sensitive to the choice of $\alpha$, and $\alpha = 0.5$ leads to the most robust performance. As noted in Appendix B.2, we found $\alpha = 0.5$ to be the best value for SAC+DIAYN after evaluating its performance for a single seed on one of the obstacle perturbation environments, and we therefore used this value of $\alpha$ for our experiments in Section 7.

Figure 6: On HalfCheetah-Goal, we study the effects on performance when (a) varying $\epsilon$ in the obstacle test environments, (b) varying $\epsilon$ in the force test environments, (c) varying $\alpha$ in the obstacle test environments, and (d) varying $\alpha$ in the force test environments. In (a) and (b), $\alpha = 1.0$, and in (c) and (d), $\epsilon = 0.1R$, where $R$ is the return achieved by a trained SAC policy on the training environment. For a single seed, we plot the mean performance over 5 runs of the best latent-conditioned policy on each test environment. SMERL is more sensitive to hyperparameter settings in the obstacles test environments as compared to the force test environments.

Figure 7: On the Obstacle and Force test environments for HalfCheetah-Goal and WalkerVelocity, we study the effect of varying $\alpha$, the weight by which the unsupervised reward is multiplied, on the evaluation performance of SAC+DIAYN. For a single seed, we plot the mean performance over 5 runs for the best latent-conditioned policy. We find that $\alpha = 0.5$ leads to the most robust performance across varying degrees of perturbations to the training environment.

## B.4   SMERL Policy Selection

We report the performance achieved by all SMERL policies on a subset of the obstacle and force test environments (see Tables 4 - 9). We also report which policy is selected by SMERL on each of the test environments. The results reported are for a single seed. We find that different SMERL policies are optimal for different degrees of perturbation to the training environment (with the exception of the HalfCheetahGoal + Obstacle test environments). In particular, the best performing policy on the train environment is not necessarily the best policy on the test environments. Further, policy selection may differ between different types of test perturbations. For example, on HalfCheetah-Goal, policy 5 is consistently the best for varying obstacle heights, whereas policies 1, 2, and 3 are sometimes better than policy 5 on the force perturbation test environments.

## B.5   When does SMERL Fail?

SMERL works well on test environments for which the test environment's optimal policy is only slightly sub-optimal on the train environment, as described in Definition 1. This assumption will not hold in all real-world problem settings, and in those settings, SMERL will not be robust to test environments. However, we expect this assumption to hold in settings where an environment changes locally (i.e. a few nearby states) and there is another path that is near optimal. This is often true in real robot navigation and manipulation problems when there are a small number of new obstacles or local terrain changes. We also expect it to be true when there is a large action space (e.g. recommender systems) and local perturbations (e.g. changes in the content of a small number of items).

In the continuous control experiments in Section 7, we found that when the degree of perturbation increases in a test environment relative to the train environment (e.g. obstacle height, force magnitude, number of time steps for which motor failure occurs), SMERL's performance decreases. This result occurs because the difference between the train environment's optimal return $R_{\mathcal{M}}(\pi^*_{\mathcal{M}})$ and the return

| Obstacle Height | Policy 1 | Policy 2 | Policy 3 | Policy 4 | Policy 5 |
|---|---|---|---|---|---|
| 0.0 | -86.3 | -87.2 | -133.1 | -77.0 | **-72.3** |
| 0.1 | -79.4 | -80.5 | -79.1 | -78.6 | **-74.3** |
| 0.2 | -76.8 | -81.7 | -71.9 | -80.1 | **-69.5** |
| 0.3 | -78.7 | -88.0 | -72.9 | -81.6 | **-68** |
| 0.4 | -85.9 | -87.7 | -88.2 | -74.0 | **-69.7** |
| 0.5 | -83.2 | -87.3 | -72.9 | -78.2 | **-68.5** |

Table 4: SMERL policy performance and selection on HalfCheetah-Goal+Obstacle test environments.

| Force Magnitude | Policy 1 | Policy 2 | Policy 3 | Policy 4 | Policy 5 |
|---|---|---|---|---|---|
| 0.0 | -86.3 | -87.2 | -133.1 | -77.0 | **-72.3** |
| 100.0 | -88.9 | -92.8 | -87.5 | -107.8 | **-83.8** |
| 300.0 | **-222.7** | -357.0 | -397.9 | -1238.7 | -424.1 |
| 500.0 | -868.9 | -528.3 | **-283.7** | -1196.3 | -669.5 |
| 700.0 | -1046.6 | -951.5 | -769.7 | -1758.8 | **-913.9** |
| 900.0 | -1249.3 | **-1238.3** | -1425.1 | -1264.4 | -1282.5 |

Table 5: SMERL policy performance and selection on HalfCheetah-Goal+Force test environments.

| Obstacle Height | Policy 1 | Policy 2 | Policy 3 | Policy 4 | Policy 5 |
|---|---|---|---|---|---|
| 0.0 | **894.2** | 854.6 | 793.8 | 853.5 | 856.3 |
| 0.1 | **891.6** | 648.6 | 659.0 | 830.5 | 693.2 |
| 0.2 | 688.8 | 439.9 | 640.0 | **764.8** | 600.1 |
| 0.3 | **766.3** | 513.8 | 452.5 | 701.2 | 464.2 |
| 0.4 | 500.4 | 459.9 | 576.0 | **613.5** | 463.7 |
| 0.5 | 454.1 | 434.9 | **479.3** | 434.5 | 462.1 |

Table 6: SMERL policy performance and selection on WalkerVelocity+Obstacle test environments.

| Force Magnitude | Policy 1 | Policy 2 | Policy 3 | Policy 4 | Policy 5 |
|---|---|---|---|---|---|
| 0.0 | **894.2** | 854.6 | 793.8 | 853.5 | 856.3 |
| 10.0 | **890.7** | 850.3 | 825.3 | 874.5 | 849.8 |
| 30.0 | **884.0** | 810.1 | 383.5 | 243.4 | 837.5 |
| 50.0 | **787.6** | 19.4 | 380.9 | 656.1 | 185.0 |
| 70.0 | 474.8 | **513.9** | 27.6 | 54.7 | 6.4 |
| 90.0 | 34.6 | 39.0 | 30.0 | **122.2** | 2.7 |

Table 7: SMERL policy performance and selection on WalkerVelocity+Force test environments.

Figure 8: The relationship between sub-optimality ($R_{\mathcal{M}}(\pi^*_{\mathcal{M}}) - R_{\mathcal{M}}(\pi^*_{\mathcal{M}'})$) and SMERL's performance on the Walker-Velocity + Force test environments.

| Obstacle Height | Policy 1 | Policy 2 | Policy 3 | Policy 4 | Policy 5 |
|---|---|---|---|---|---|
| 0.0 | **663.9** | 658.0 | 612.6 | 630.0 | 630.8 |
| 0.1 | **661.7** | 652.6 | 612.1 | 630.8 | 630.6 |
| 0.2 | **666.2** | 653.8 | 610.9 | 630.2 | 631.4 |
| 0.3 | 514.1 | 604.6 | 608.4 | **631.5** | 600.7 |
| 0.4 | 535.9 | 556.7 | 539.4 | **557.4** | 552.8 |
| 0.5 | 528.3 | **562.3** | 539.1 | 552.5 | 546.7 |

Table 8: SMERL policy performance and selection on HopperVelocity-Goal+Obstacle test environments.

| Force Magnitude | Policy 1 | Policy 2 | Policy 3 | Policy 4 | Policy 5 |
|---|---|---|---|---|---|
| 0.0 | **663.9** | 658.0 | 612.6 | 630.0 | 630.8 |
| 10.0 | **667.9** | 655.5 | 611.5 | 629.7 | 629.7 |
| 30.0 | **671.7** | 633.5 | 614.1 | 626.3 | 630.3 |
| 50.0 | **667.5** | 617.6 | 602.4 | 622.6 | 632.1 |
| 70.0 | 376.3 | 585.0 | **617.9** | 612.4 | 630.0 |
| 90.0 | 191.0 | 547.1 | **632.8** | 605.4 | 635.2 |

Table 9: SMERL policy performance and selection on HopperVelocity-Goal+Force test environments.

achieved by the test environment's optimal policy on the train environment $R_{\mathcal{M}}(\pi^*_{\mathcal{M}'})$ increases as the perturbation amount increases. We verify this experimentally by comparing $R_{\mathcal{M}}(\pi^*_{\mathcal{M}}) - R_{\mathcal{M}}(\pi^*_{\mathcal{M}'})$ to SMERL's return on the Walker-Velocity + Force test environments (see Figure 8). Concretely, the train MDP $\mathcal{M}$ is Walker-Velocity with no force applied, and the test MDPs $\mathcal{M}'$ have various magnitudes of force applied. We find that as $R_{\mathcal{M}}(\pi^*_{\mathcal{M}}) - R_{\mathcal{M}}(\pi^*_{\mathcal{M}'})$ increases, SMERL's performance on the corresponding test MDP $\mathcal{M}'$ decreases.