[Reviews · NeurIPS 2020]

Review 1

Summary and Contributions: The paper proposes an approach to learn diverse behaviors to avoid policies to be too specific for a single task and making them general and robust to variations of the task. The proposed method considers policies depending on latent variables and optimizes an objective that prefers policies with high mutual information between the trajectory and the latent variable, conditioned to the fact that those policies must be \epsilon-optimal. Differently from meta-learning, the training is carried out in a single environment while testing is done on variations. A theoretical study to justify the proposed objective is provided together with an experimental evaluation.

Strengths: The main strength point of the paper is the attempt of addressing a quite challenging scenario in which training can be carried out in a single environment while testing should be done in different environments.

Weaknesses: - The objective function (1): I have two concerns about the definition of this objective: 1. If the intuitive goal consists of finding a set of policies that contains an optimal policy for every test MDP in S_{test}, I would rather evaluate the quality of \overline{\Pi} with the performance in the worst MDP. In other words, I would have employed the \min over S_{test} rather than the summation. With the summation we might select a subset of policies that are very good for the majority of the MDPs in S_{test} but very bad of the remaining ones and this phenomenon would be hidden by the summation but highlighted by the \min. 2. If no conditions on the complexity of \overline{\Pi} are enforced the optimum of (1) would be exactly \Pi, or, at least, the largest subset allowed. - Latent-Conditioned Policies: How is this different from considering a hyperpolicy that is used to sample the parameters of the policy, Like in Parameter-based Exploration Policy Gradients (PGPE)? Sehnke, Frank, et al. "Policy gradients with parameter-based exploration for control." International Conference on Artificial Neural Networks. Springer, Berlin, Heidelberg, 2008. - Choice of the latent variable distribution: at line 139 the authors say that p(Z) is chosen to be the uniform distribution, while at line 150 p(Z) is a categorical distribution. Which one is actually used in the algorithm? Is there a justification for choosing one distribution rather than another? Can the authors motivate? **Minor*** - line 64: remove comma after a_i - line 66: missing expectation around the summation - line 138: what is H(Z)? - line 322: don’t -> do not - Equation (2): at this point the policy becomes a parametric function of \theta. Moreover, the dependence on s when using the policy as an argument for R_{\mathcal{M}} should be removed - Figure 3: the labels on the axis are way too small - Font size of the captions should be the same as the text

Correctness: The proofs of all claims are reported in appendix. I made a high-level check of the math and seems correct to me.

Clarity: The paper is clearly written.

Relation to Prior Work: The proposed approach shares similarities with metalearning and robust RL. The connections are appropriately discussed in Section 6.

Reproducibility: Yes

Additional Feedback: The paper has nice potential. Considering my concerns about the definition of the objective function, at present, I am not sure that this paper represents a sufficient contribution to be suited for publication at NeurIPS. ***Post Rebuttal*** I thank the authors for the feedback. I have read it together with the other reviews. I am happy that the authors clarified my issue about the objective function. I think that the paper, provided that the authors make the promised fixes, is a suitable contribution for NeurIPS. For this reason, I increase my score to 6.


Review 2

Summary and Contributions: UPDATE after rebuttal I thank the authors for providing more intuition about the settings in which the method can be expected to succeed or fail. I hope they will follow through with the promise to include additional experiments that emphasize the limitations of this approach, as well as more in-depth analysis of the learned policies, as suggested in my review. I believe these additions would be valuable for readers and would improve the paper overall. However, due to the somewhat limited applicability and novelty of the proposed method, I will keep my score. ---- This paper proposes a new approach for generalizing to a certain type of out-of-distribution environments by training a diverse set of near-optimal policies on a single training environment. They formalize the family of environments to which this method is robust as a function of the training environment and its optimal policy. The method outperforms a series of baselines on a wide range of environments with new transition and / or reward functions.

Strengths: I liked this paper overall. The problem formulation and method description are clear and easy to read. I also appreciated that the paper includes both some theoretical analysis and empirical validation. The problem of generalizing to out-of-distribution environments using only a few episodes is an important one. The proposed approach is also novel as far as I know, even if it builds on ideas from prior work. Another strength of this paper is the precise description of the test environments on which you can expect the proposed approach to be robust to.

Weaknesses: My main concern about this paper is the generality and effectiveness of this method on more complex and realistic settings. It seems to me that the environments used for evaluating the method were specifically designed in such a way that the proposed method can work well. I am not sure how realistic this assumption (that near-optimal policies on the train environments will be near-optimal for the test environments) is for more real-world problems we might want to solve. My understanding is that this method would only work well for test environments that are still fairly similar to the training one. So it is not clear how much you gain by doing this rather than doing domain randomization or adversarial robustness training. It would be useful to provide more analysis of how the method works in practice. For example, it would be valuable to report the performance achieved by all the policies on both the training environments and the test environments, as well as the correlation between the training and test performance for a given policy. This would provide more understanding about how how much variance there is in the performance of the different policies as well as the difference in successful behaviors between train and test. It would also be useful to report which policy is selected by the algorithm as being the best for each of the test environments as well as how far its performance is from the optimal policy on that environment (e.g. one directly trained on the test environment using SAC). This would be a good sanity check to ensure that there isn't a single policy (perhaps the optimal training one) that always performs bets on the train environments.

Correctness: Yes

Clarity: Yes

Relation to Prior Work: Yes

Reproducibility: Yes

Additional Feedback:


Review 3

Summary and Contributions: The paper proposes a new algorithm which objective is to learn a set of policies that are robust to variations of the training environment at test time (while trained on a single and fixed MDP). The principle is to learn a latent-condition policy (the latent variable being used to encode a set of different policies) such that, for each value of the latent variable, the resulting policy achieves a reward close to the reward of the optimal policy in the training MDP, but is different from the policies encoded through other latent variables values. Concretely, it is done by extending the 'Diversity is all you need' (DIAYN) algorithm, considering a task reward in addition to the diversity driven reward. Instead of using a naive combination of the two criteria (which is done as a baseline in the experimental section), the authors describe a threshold-based approach such that the diversity is optimized only if a critical reward is obtained by the policy. In addition to a concrete algorithm, the paper provides a study of the properties of such an approach to understanding in which case this model is useful, and which types of modifications in the original MDOP it can deal with. Experiments are made on simple and classical environments, with a comparison to multiple baselines showing the interest of the approach.

Strengths: + The paper targets an important problem which is usually solved by using a different training setting where multiple variations of an environment are available at train time. In the article case, it is interesting to see that the method works training only one a simple environment. It is thus usable in applications where interaction between the world and the agent is expensive + The method is a modification of DIAYN that may be simple to reproduce (See next section) and can be plugged in any RL algorithm. It is provided with an interesting analysis of the properties of the produced policies. + Experiments are convincing, even if made on simple environments + subjective opinion: I am convinced that training multiple policies instead of a single one is a very nice approach to many aspects of RL and I am happy to see more and more paper proposing concrete ways to do it, and concrete use-cases. Such a paper will interest a large audience.

Weaknesses: Some aspects of the method are not clear. My main concern is on the computation of equation (4)/eq (5). The notation R_M refers to the reward obtained by a particular policy and the idea is to consider the diversity reward only if the current policy is able to achieve a reward that is greater than the optimal reward minus epsilon. What is not clear to me is how this optimal reward is computed: is it the expectation of the future reward obtained in the current state by the optimal policy ? or is it just a comparison in term of reward over the whole trajectories of the two policies starting from the same state? If yes, how do you compute this difference when different initial states are sampled? Do you maintain an average reward value for each of the trained policies? In terms of practical implementation, is it just a reward modification, or does it have consequences over the advantage estimation/critic? If you compare the current policy and the optimal policy at each state, how do you manage the fact that these two policies are not generating states with the same distribution (such that the estimated value for the optimal policy may not be accurate), etc.? So I have many concerns on that point and can imagine many different ways to implement the method. Which one is the right one? Having an explicit pseudo-code for the SMERL (with SAC) algorithm would help. In terms of experiments, problems with a high dimensional input space (e.g pixels) are lacking (but it is not crucial) and would strengthen the paper. The best would be to have a 'real' use-case instead of hand-made ones, for instance in robotics, moving in a house where furniture may move between two episodes. The way epsilon is chosen is not clear. How do you cross-validate on espilon if you don't have validation MDPs. Similarly, how do you choose the right number of policies to learn ? How about having a continuous latent variable instead of a discrete one ? Having a small discussion on that point would be interesting. The fact that SAC+DIAYN results do not appear in the paper (but in the supplementary material) is strange since it is an important comparison. Moreover, I would be curious to understand how the SAC+DIAYN is concretely done (through a weighted sum? how do you choose the value of the weight in that case) since it may certainly also be a good approach when carefully tuned

Correctness: Everything seems correct, not particular comments on that point.

Clarity: The paper is well written, and well structured.

Relation to Prior Work: Multiple connections are made with the existing related work. Since the proposed model is a simple extension of DIAYN, I would (again) move the SAC+DIAYN model from the supplementary material to the main article, and better describe the differences.

Reproducibility: No

Additional Feedback: Reproducibility is not easy (see my comments) and providing a pseudo-code showing the concrete implementation would help.


Review 4

Summary and Contributions: This paper presents an algorithm that learns diverse ways to solve a given task. The policy is conditioned on the latent variable, and the lower bound of the mutual information between the state and the latent variable is maximized. The lower bound of the mutual information is based on the approach used in DIAYN. The benefits of learning diverse solutions are shown as robustness to perturbations to a task.

Strengths: Learning diverse solutions is an interesting research direction, and it is not trivial to balance the objective that encourage the diversity of solutions and the objective to solve a given task. The way of encourage the diversity of solution while keeping the solution quality may inspire the researchers in the NeurIPS community.

Weaknesses: To balance the diversity of solutions and the objective to solve a given task, the proposed method introduced the constrained information maximization in (2). However, the proposed algorithm needs to know the optimal return R_M(\pi^*_M). Therefore, before using the proposed method, it is necessary to solve the given task using another off-the-shelf RL method.

Correctness: In the experiment, the training condition for Robust Adversarial RL is not clear. I think that the robustness of the policy learned with RARL is dependent on the hyperparameters of the adversarial. The authors need to describe the training conditions for RARL and clarify how they are determined.

Clarity: The paper is overall clearly written and it is easy to follow except some parts. I’m not sure how the latent variable z is sampled during the learning process. Although it seems that the latent variable is fixed during an episode in the adaptation phase, I’m not sure about the learning phase.

Relation to Prior Work: Learning a policy conditioned on the latent variable for diverse behaviors appears in the context of imitation learning as well. I think the following studies should be also cited and discussed in the related work section. [1] Y. Li, J. Song, S. Ermon. InfoGAIL: Interpretable Imitation Learning from Visual Demonstrations. NeurIPS 2017. [2] Mere et al., Neural Probabilistic Motor Primitives for Humanoid Control, ICLR 2019 Regarding the diverse trajectories for achieving a task, the following studies on hierarchical policy search seems relevant (See Figure 3 in [3]). This study addresses the special type of HRL in which an option is selected in the beginning of the episode. (See Figure 3, which shows the two paths to reach the specified goal.) [3] Christian Daniel, Gerhard Neumann, Oliver Kroemer, Jan Peters; Hierarchical Relative Entropy Policy Search, Journal of Machine Learning Research, 17(93):1−50, 2016.

Reproducibility: No

Additional Feedback: - In the supplementary material, I do not clearly understand how the hyperparameter B is used. I think that B does not appear in the main text. Please elaborate it. - I’m not sure why the results of SAC + DIAYN is only in the supplementary. I think it should be in the main text since it would add important information. === comments after rebuttal === I have read the other reviews and author response. I appreciate the authors' efforts to answer the questions raised by reviewers. Although the author response clarified some points, I did not find new information that makes me increase the score. I keep the initial score.

[Author Response · NeurIPS 2020]

We thank the reviewers for the constructive feedback. We will incorporate the valuable suggestions in the revised
version. We included prior works suggested by **R1** and **R5** in Section 6 and moved SAC+DIAYN results to the main
text (**R4** and **R5**).

**R1: Worst case objective in Equation 1,** min **over** $S_{\text{test}}$ **vs summation.** Thank you for catching this important typo.
The summation should actually be a $\min$, and we will fix this in the revised paper. The motivation in Section 4.1 above
Eq 1 and all references and theoretical results using the equation are consistent with a $\min$, rather than a summation.

**R1: Complexity of** $\bar{\Pi}$. As noted in Proposition 4's proof, SMERL finds an optimal policy *for each MDP in* $S_{\text{test}}$, if
$|\bar{\Pi}| \geq |S_{\pi_{\mathcal{M}}^*,\varepsilon}|$. The set $S_{\pi_{\mathcal{M}}^*,\varepsilon}$ is generally going to be much smaller than $\Pi$, so we can choose a size of $\bar{\Pi}$ that is much
smaller than $|\Pi|$. However, $|S_{\pi_{\mathcal{M}}^*,\varepsilon}|$ is still large, and as we note in Remark 2, metric space smoothness assumptions on
MDPs in $S_{\text{test}}$ are needed to to obtain reasonable, non worst-case performance guarantees *for all* test MDPs when a
smaller $\bar{\Pi}$ is used. A natural next step is to perform an analysis with smoothness assumptions, and we will include this
analysis in the final version. In our practical SMERL algorithm that we use for our experiments, $\bar{\Pi}$ is restricted to only
contain 5 policies, so the optimum $\bar{\Pi}$ is much smaller than $\Pi$.

**R1: SMERL vs hyperpolicy (e.g., PGPE).** A key consequence of this prior approach is that the discriminator in
SMERL would need to predict the policy parameters, which is more difficult than predicting a low-dimensional latent
variable in a latent-variable policy. Additionally, the PGPE paper does not consider the problem of few-shot robustness.

**R1: Choice of latent distribution.** $p(Z)$ is the uniform distribution over the space of latent variables. In our
experiments, $Z$ is a uniform random variable over a discrete set of values. We choose $p(Z)$ to be uniform to maximize
the entropy $H(Z)$ over the latent variables, which is needed to maximize $I(s_t; z)$ in Equation 3.

**R3: Environments where SMERL is expected to succeed.** We will add a discussion of settings in which we expect
SMERL to not succeed and add an experimental setup in which we expect SMERL to fail, for example by making a
perturbation non-local (e.g. many obstacles rather than a single obstacle).

**R3: Assumption: near-optimal policies on the train MDPs will be near-optimal for the test environments.** We
agree that this assumption will not hold in all real-world problem settings. We expect this to hold in settings where
an environment changes locally (i.e. a few nearby states) and there is another path that is near optimal. This is often
true in real robot navigation and manipulation problems when there are a small number of new obstacles or local
terrain changes. We also expect it to be true when there is a large action space (e.g. recommender systems) and local
perturbations (e.g. changes in the content of a small number of items).

**R3: Report performance of all the SMERL policies...report which policy is selected.** We added a brief sum-
mary of these results on the HalfCheetah-Goal + Obstacles perturbation environment in the following table. We
will add a more detailed empirical analysis of policy performance and selection by SMERL to the revised paper.

| Obstacle | Policy ID | $(z_i)$ | | |
|---|---|---|---|---|
| Height | 0 | 1 | 2 | 3 |
| 0.0 | **-72.2** | -87.2 | -86.1 | -107.9 |
| 0.15 | **-72.2** | -87.2 | -86.1 | -107.9 |
| 0.3 | -653.2 | -721.1 | **-86.1** | -1004.3 |
| 0.45 | -690.1 | -761.6 | **-86.1** | -1004.3 |

**R4: Optimal reward computation.** The optimal return is the expected discounted sum of rewards obtained over an entire episode by the optimal policy. In practice, we estimate the optimal return by measuring the return achieved by a trained SAC agent's policy over the whole trajectory averaged over different initial states. In SMERL, we compare the current policy's whole trajectory return with the optimal return.

39  **R4: Difference between the optimal return and the current policy's return with different initial states.** In the
40  experiments, the variability in the initial state distribution was small, so the SMERL policy's trajectory return for each
41  episode was individually compared with the optimal return estimate. In other environments, a separate optimal return
42  estimate for different initial state regions may be required.

43  **R4: Selection of** $\varepsilon$. We experimented with different values of $\varepsilon$ on the HalfCheetah-Goal + Obstacle task (see Fig. 5 in
44  App. B). We computed the percentage of the optimal return that this $\varepsilon$ value corresponds to, and used this percentage to
45  select $\varepsilon$ for all environments. We have added a description of this protocol to the revised paper.

46  **R4: SAC+DIAYN.** SAC+DIAYN uses a weighted sum of the environment and DIAYN reward. We experimented with
47  3 different weights on the DIAYN reward on the HalfCheetah-Goal + Obstacle test environment and found SMERL to
48  consistently outperform SAC+DIAYN. We have included these results in the revised paper.

49  **R5: RARL.** For RARL, we performed a grid search on HalfCheetah-Goal with evaluation on the Obstacles perturbation
50  to select the hyper-parameters of TRPO, the policy optimizer for the protagonist and adversarial policies. We then kept
51  these hyper-parameters fixed for all environments. We have added a detailed description to the revised paper.

52  **R5: How are latent variables sampled?** During both the learning (see Alg 1) and the adaptation phases, the latent
53  variable $z$ is sampled at the beginning of an episode and is held fixed for the episode. We will clarify this Section 4.2.

54  **R5: How is the hyper-parameter B used?** The hyper-parameter $B$ is associated with epsilon in Equations 2-5. It is
55  the return value that SMERL must achieve prior to the unsupervised reward being added to the objective. Specifically,
56  $B = R_{\mathcal{M}}(\pi_{\mathcal{M}}^*) - \varepsilon$. We have added this description to the revised paper.

[Meta-Review · NeurIPS 2020]

The paper studies the interesting problem of generalization after a single task training. The idea and algorithmic development are well grounded and they lead to an algorithm that has good empirical performance against baselines. In order to improve the submission further, the authors may have a more explicit comparison of the algorithmic differences between their approach and SAC+DIAYN.